# Nuclear Expression of TDP-43 Is Linked with Morphology and Ubiquitylation of Cytoplasmic Aggregates in Amyotrophic Lateral Sclerosis

**DOI:** 10.3390/ijms241512176

**Published:** 2023-07-29

**Authors:** Hiroyuki Yabata, Yuichi Riku, Hiroaki Miyahara, Akio Akagi, Jun Sone, Makoto Urushitani, Mari Yoshida, Yasushi Iwasaki

**Affiliations:** 1Department of Neuropathology, Institute for Medical Science of Aging, Aichi Medical University, Nagakute 480-1195, Aichi, Japan; yabata@belle.shiga-med.ac.jp (H.Y.); miyahara.hiroaki.926@mail.aichi-med-u.ac.jp (H.M.); akio.akagi@gmail.com (A.A.); jsone@aichi-med-u.ac.jp (J.S.); myoshida@aichi-med-u.ac.jp (M.Y.); iwasaki@sc4.so-net.ne.jp (Y.I.); 2Department of Neurology, Shiga University of Medical Science, Otsu 520-2192, Shiga, Japan; uru@belle.shiga-med.ac.jp; 3Department of Neurology, Nagoya University, Nagoya 466-8550, Aichi, Japan

**Keywords:** ALS, autopsy, FTLD, mislocalization, motor neuron, spinal cord, TDP-43, and ubiquitin

## Abstract

The transactive response DNA-binding protein of 43 kDa (TDP-43) is a pathological protein of amyotrophic lateral sclerosis (ALS). TDP-43 pathology is characterized by a combination of the cytoplasmic aggregation and nuclear clearance of this protein. However, the mechanisms underlying TDP-43 pathology have not been fully clarified. The aim of this study was to evaluate the relationships between the expression level of nuclear TDP-43 and the pathological properties of cytoplasmic aggregates in autopsied ALS cases. We included 22 consecutively autopsied cases with sporadic TDP-43-related ALS. The motor neuron systems were neuropathologically assessed. We identified 790 neurons with cytoplasmic TDP-43 inclusions from the lower motor neuron system of included cases. Nuclear TDP-43 disappeared in 84% (*n* = 660) and expressed in 16% (*n* = 130) of neurons with cytoplasmic inclusions; the former was defined as TDP-43 cytoplasmic immunoreactivity (c-ir), and the latter was defined as nuclear and cytoplasmic immunoreactivity (n/c-ir). Morphologically, diffuse cytoplasmic inclusions were significantly more prevalent in TDP-43 n/c-ir neurons than in c-ir neurons, while skein-like and round inclusions were less prevalent in n/c-ir neurons. The cytoplasmic inclusions of TDP-43 n/c-ir neurons were phosphorylated but poorly ubiquitylated when compared with those of c-ir neurons. TDP-43 n/c-ir neurons became less dominant than the c-ir neurons among cases with a prolonged disease duration. The expression level of nuclear TDP-43 was significantly lower in n/c-ir neurons than in normal neurons without cytoplasmic inclusions. Our results indicate that the maturation of cytoplasmic TDP-43 inclusions correlates with the depletion of nuclear TDP-43 in each affected neuron. This finding supports the view that an imbalance between nuclear and cytoplasmic TDP-43 may be an essential pathway to TDP-43 pathology.

## 1. Introduction

Amyotrophic lateral sclerosis (ALS) is a progressive neurodegenerative disorder that involves the upper and lower motor neuron systems. Clinical findings of ALS are characterized by a combination of upper and lower motor neuron signs. The pathological characteristics of ALS comprise the loss of upper and lower motor neurons in association with protein aggregation within neurons and glia. The transactive response DNA-binding protein of 43 kDa (TDP-43) is known as a pathological protein of ALS, and approximately 90% of autopsied ALS cases have TDP-43 aggregates [1,2,3]. Cytoplasmic aggregates of TDP-43 typically appear to be skein-like, round, or diffuse inclusions, which are phosphorylated and ubiquitylated [4]. TDP-43 is also known as a major pathological protein of frontotemporal lobar degeneration (FTLD); approximately 50% of FTLD cases have TDP-43 aggregates [5]. ALS and FTLD are often concurrent, and this phenotype is termed frontotemporal dementia with motor neuron disease (FTD-MND) or ALS with dementia. TDP-43-related ALS (ALS-TDP) and FTLD (FTLD-TDP) can be collectively included in a disease spectrum, ‘TDP-43 proteinopathies’ [6,7].

TDP-43 is an RNA-binding protein that is localized within the nucleus in normal settings [8]. In affected neurons of ALS-TDP or FTLD-TDP cases, TDP-43 aggregates within the cytoplasm or neurites and disappears from the nucleus [1,2]; the term ‘TDP-43 pathology’ is usually defined by the combination of cytoplasmic aggregation and nuclear clearance. Hence, the pathogenesis of TDP-43 pathology may comprise two factors: (1) a gain of neurotoxicity from cytoplasmic aggregates, and (2) a loss of TDP-43 functions [9]. For instance, it has been reported that the aggregation of misfolded TDP-43 results in cell death [10]. Moreover, misfolded TDP-43 functions as a seed for further aggregation and has been hypothesized to propagate along neural connectivity [11,12,13] or via extracellular secretion [14,15]. Although the functions of intrinsic TDP-43 remain to be elucidated, its contributions to DNA repair, neuronal growth, and synaptic plasticity have been reported [16,17,18]. A reduction in nuclear TDP-43 has been reported to result in motor neuron death, astrogliosis, and an ALS-like phenotype [19,20]. However, the mechanism of TDP-43 pathology remains to be elucidated. A combination of the aggregation and nuclear clearance of TDP-43 in neurons has been shown in familial ALS cases with *TDP-43* gene mutation or aggregation models induced by patient-derived TDP-43 seed [21,22]. This fact indicates that mutation-derived or seed-induced structural alterations of TDP-43 may facilitate not only the aggregation but also the mislocalization of nuclear TDP-43. In contrast, previous studies have occasionally shown that the cytoplasmic aggregation and nuclear clearance of TDP-43 are not always coupled together. For instance, some postmortem studies of ALS cases display the presence of motor neurons with cytoplasmic inclusions and nuclear expression of TDP-43 [23,24]. It has also been reported that some aggregation models of TDP-43 demonstrate cytoplasmic aggregation of ubiquitylated TDP-43 but not nuclear clearance [25].

The aim of this study was to clarify whether the depletion of nuclear TDP-43 is linked to the properties of TDP-43 cytoplasmic aggregates in autopsied cases with ALS. We undertook the research with the following processes: first, we identified neurons with cytoplasmic aggregates in combination with nuclear clearance of TDP-43 and those with cytoplasmic aggregates in presence of nuclear TDP-43 immunoreactivity; second, we compared the morphology and ubiquitylation of cytoplasmic TDP-43 aggregates between these groups of neurons. Consecutive autopsied cases with sporadic ALS-TDP were studied.

## 2. Results

### 2.1. Cases

The demographics of the included cases are summarized in Table 1. We included 22 consecutively autopsied cases with sporadic ALS-TDP (8 females and 14 males). The median disease duration was 19 months, ranging from 6 to 132 months. A case who survived for 12 months underwent artificial ventilation during the last three weeks. Two cases had the FTD-MND phenotype in association with FTLD-TDP type B. We also included age-matched 15 controls without neurological disorders (3 females and 12 males). Postmortem delay, the severity of aging-related pathology in the brain, and brain weight did not differ between the ALS cases and controls.

#### 2.1.1. TDP-43 Pathology in the Lower Motor Neuron System

The hypoglossal nerve nucleus and spinal cord were subjected to anti-full length TDP-43 (panTDP-43) immunohistochemistry that was able to immunolabel both cytoplasmic aggregates and intrinsic nuclear TDP-43. We defined cytoplasmic TDP-43 immunoreactivity in combination with nuclear clearance as ‘TDP-43 c-ir (cytoplasmic immunoreactivity)’. In contrast, cytoplasmic TDP-43 immunoreactivity in combination with nuclear expression was defined as ‘TDP-43 n/c-ir (nuclear and cytoplasmic immunoreactivity)’. All 22 cases with ALS-TDP showed TDP-43 c-ir neurons in the anterior horn of the spinal cord, and 16 cases also showed TDP-43 c-ir neurons in the hypoglossal nerve nucleus. In total, 15 out of the 22 ALS-TDP cases (68%) had TDP-43 n/c-ir neurons in the spinal cord anterior horn, and 5 of them also had n/c-ir neurons in the hypoglossal nerve nucleus (Figure 1). In contrast, the 15 age-matched controls without neurodegenerative disorders did not show any neuronal TDP-43 aggregates in the spinal cord or hypoglossal nerve nucleus.

#### 2.1.2. Morphological Features of Cytoplasmic TDP-43 Inclusions Differed between TDP-43 n/c-ir and TDP-43 c-ir Neurons

We identified 790 neurons with cytoplasmic TDP-43 aggregates from the lower motor neuron systems of the studied cases, with a total of 130 neurons with TDP-43 n/c-ir and 660 neurons with TDP-43 c-ir (Appendix A). Cytoplasmic aggregates of TDP-43 n/c-ir neurons were found to be diffuse, skein-like, or round inclusions, similar to those of TDP-43 c-ir neurons. However, the prevalence of each inclusion subtype significantly differed between TDP-43 n/c-ir and TDP-43 c-ir neurons (Figure 2A). Diffuse inclusions were more prevalent in TDP-43 n/c-ir neurons than in TDP-43 c-ir neurons (*p* < 0.00001, Fisher’s exact test, significance level was set at 0.016 after Bonferroni correction), whereas skein-like and round inclusions were more prevalent in TDP-43 c-ir neurons than in n/c-ir neurons (skein-like inclusions *p* = 0.0142; round inclusions *p* = 0.0002). In particular, round inclusions were very rare in TDP-43 n/c-ir neurons.

#### 2.1.3. Cytoplasmic TDP-43 Inclusions of n/c-ir Neurons Were Phosphorylated but Poorly Ubiquitylated

Double immunofluorescence toward panTDP-43 and phosphorylated TDP-43 (p-TDP-43) revealed that the cytoplasmic TDP-43 component of n/c-ir neurons was phosphorylated, whereas the nuclear TDP-43 component was not (Figure 3). Double immunofluorescence toward panTDP-43 combined with ubiquitin or p-62 clarified that the cytoplasmic TDP-43 of n/c-ir neurons was ubiquitylated and p62-tagged (Figure 3). However, colocalization between cytoplasmic inclusions and ubiquitin or p-62 tended to be poorer in TDP-43 n/c-ir neurons than in TDP-43 c-ir neurons. Cytoplasmic TDP-43 aggregates of c-ir neurons almost completely colocalized with ubiquitin and p-62, but the colocalization remained segmental in n/c-ir neurons (Figure 3). We subjected three cases with TDP-43 n/c-ir neurons to a colocalization assay for TDP-43 and ubiquitin; ten TDP-43 n/c-ir and ten TDP-43 c-ir neurons of the cervical cord were assessed. We evaluated neurons with skein-like inclusions in this assay. The assay demonstrated a significantly lower colocalization rate in TDP-43 n/c-ir neurons than in c-ir neurons (*p* = 0.0003, Mann-Whitney U test) (Figure 3).

#### 2.1.4. Nuclear TDP-43 Signal Intensity Was Lower in n/c-ir Neurons Than in Normal Neurons

Nuclear immunostaining of TDP-43 tended to be weaker in TDP-43 n/c-ir neurons than in normal neurons that did not have any TDP-43 inclusions. We subjected three cases with TDP-43 n/c-ir neurons to a signal intensity assay using anti-panTDP-43 immunofluorescence. Ten TDP-43 n/c-ir neurons, ten TDP-43 c-ir neurons, and ten normal neurons from the cervical cord were assessed for each case. The assay clarified a significantly lower signal intensity of nuclear TDP-43 in n/c-ir neurons than in normal neurons (*p* < 0.00001, Mann-Whitney U test) (Figure 4 and Figure 5).

### 2.2. TDP-43 n/c-ir Neurons in the Upper Motor Neuron System

In total, 16 out of the 22 studied ALS cases had neuronal TDP-43 aggregates in the primary motor cortex, and 4 of them showed TDP-43 n/c-ir. TDP-43 n/c-ir was observed in small pyramidal neurons and occasionally in Betz cells (Figure 1). These four cases had TDP-43 n/c-ir in the lower motor neurons, and one of these cases had the FTD-MND phenotype.

### 2.3. TDP-43 n/c-ir Glia in the Motor Neuron System

It is known that glia are often involved in TDP-43 pathology of ALS cases [26]. Glial TDP-43 aggregates are mostly oligodendroglial in origin [26], which typically show inclusions within the cytoplasm or myelin sheath in combination with nuclear clearance [27]. All 22 studied cases with ALS showed glial cytoplasmic inclusions in the spinal cord, hypoglossal nerve nucleus, and primary motor cortex. In total, 13 out of the 22 cases (60%) had TDP-43 n/c-ir glia in the lower motor neuron system (Figure 1), and 1 case also showed TDP-43 n/c-ir glia in the primary motor cortex.

### 2.4. TDP-43 n/c-ir Neurons Became Less Dominant with a Prolonged Disease Duration

We computed the ratio of ((TDP-43 n/c-ir neuron)/(TDP-43 c-ir neurons)) in each case to assess the dominancy of these two types of TDP-43 abnormalities in lower motor neuron systems. As mentioned above, 790 neurons with TDP-43 aggregates (130 with n/c-ir and 660 with c-ir) in total were identified from the hypoglossal nerve nucleus and spinal cord of the studied cases, and this value was used in the calculation. The ratio ((TDP-43 n/c-ir neuron counts)/(TDP-43 c-ir neuron counts)) was negatively correlated with the disease duration of ALS (rho = −0.54, *p* = 0.0095, Spearman’s rank order) (Figure 2B). The cases with a disease duration of 49 months or longer never showed TDP-43 n/c-ir neurons in the lower motor neuron system (Appendix A). Similarly, TDP-43 n/c-ir neurons of the primary motor cortex were observed in cases with clinical duration of 40 months or shorter, although *a* negative correlation between the dominance of n/c-ir neurons and clinical durations was not statistically shown (Figure 2B, Appendix A). In contrast, TDP-43 n/c-ir glia were observed in cases with both short and long disease durations, which ranged from 6 months to 132 months (Figure 2B, Appendix A).

## 3. Discussion

The study results revealed that a subset of neurons with cytoplasmic TDP-43 aggregates showed the nuclear immunoreactivity of this protein, which was defined as n/c-ir, in the motor neuron system. For lower motor neuron system, this finding was observed in more than half of the studied cases. Moreover, a subset of them had TDP-43 n/c-ir neurons in the upper motor neuron system. The morphological characteristics and ubiquitylation level of cytoplasmic TDP-43 aggregates were clearly different between n/c-ir and c-ir neurons. Our results suggest that the process of cytoplasmic TDP-43 aggregation is linked to a reduction in nuclear TDP-43.

It is known that aggregative TDP-43 is abnormally phosphorylated at several sites, and some hyperphosphorylated sites were included in C-terminal fragments [4]. We resorted immunohistochemistry toward phosphorylated serine 409 and 410 as a marker of aggregative TDP-43. Our double immunofluorescence clarified that cyto-plasmic TDP-43 in n/c-ir neurons was phosphorylated, but nuclear TDP-43 was not. This finding indicates that the cytoplasmic component of TDP-43 n/c-ir is aggregative, whereas the nuclear component is not. TDP-43 can aggregate within the neuronal nu-cleus, termed neuronal nuclear inclusions (NNIs) [28]. NNIs are usually lentiform or cat’s eye-like and frequent in the cortical neurons of FTLD-TDP type A cases. TDP-43 aggregation involving both the nucleus and cytoplasm has also been reported [24]. However, NNIs are phosphorylated, as are cytoplasmic aggregates. Hence, TDP-43 n/c-ir in our study is not a concurrence of cytoplasmic and nuclear aggregations of TDP-43. Contributions of the C-terminal hyperphosphorylation to misfolding and mislocalization of TDP-43 remain controversial. An in vitro assay suggested that ca-sein-kinase-1-mediated phosphorylation of TDP-43 facilitated filament formation, from which phosphorylated serine 409 and 410 epitopes were detected [4]. By contrast, a recent study reported that TDP-43 phase-separation and aggregation were sup-pressed by casein-kinase-1-mediated phosphorylation of TDP-43 in vitro; and C-terminal phosphomimetic mutation did not affect intranuclear localization of TDP-43 in vivo [29]. Hence, hyperphosphorylation of C-terminal is correlated with TDP-43 aggregates, but its causality toward TDP-43 pathology remains to be elucidated.

Morphological observation clarified that diffuse inclusions were most prominent in TDP-43 n/c-ir neurons, followed by skein-like inclusions; round inclusions were very rare. This is in great contrast with neurons presenting TDP-43 aggregation with nuclear clearance (c-ir neurons), in which skein-like inclusions were most prevalent. Autopsy-based study findings have suggested that diffuse cytoplasmic inclusions are poorly ubiquitylated and are presumed to represent an early phase of aggregation, termed ‘preinclusions’ [23,24]. In contrast, localized and dense inclusions, such as skein-like or round inclusions, have been considered mature inclusions, and the maturity is reported to correlate with prolonged disease duration or a reduction in neuronal counts in the anterior horn [23,30]. The dominancy of diffuse inclusions among TDP-43 n/c-ir neurons suggests that the n/c-ir represents a transitional state between the normal setting and mature TDP-43 pathology defined by nuclear clearance and cytoplasmic aggregation.

We also found that the cytoplasmic inclusions of TDP-43 n/c-ir neurons were less ubiquitylated than those of c-ir neurons. We compared only skein-like inclusions in n/c-ir and c-ir neurons to avoid an effect of subtypes of inclusions; as mentioned above, diffuse cytoplasmic inclusions are known to be poorly ubiquitylated even in neurons with nuclear clearance of TDP-43 [23]. Importantly, skein-like inclusions of TDP-43 were definitely but only partially ubiquitylated among n/c-ir neurons, whereas they were almost entirely colocalized with ubiquitin among c-ir neurons. Immunoreactivity toward p62, which is a key molecule transferring ubiquitylated protein to the autophagosomal system [31], was also less in the skein-like inclusions of TDP-43 n/c-ir neurons than in those of c-ir neurons. A previous study suggested a possibility that mislocalization of TDP-43 precedes the ubiquitylation of the inclusions because TDP-43 aggregates spread more broadly than ubiquitylated aggregates, and ubiquitin-immunopositivity corresponded to round or skein-like inclusions but not diffuse inclusions [23]. Hence, cytoplasmic TDP-43 aggregates of n/c-ir neurons may be mostly in a premature state of aggregation. Moreover, our signal intensity assay of nuclear TDP-43 indicates that the nuclear expression level of TDP-43 may partially reduce at the state of n/c-ir.

Our clinicopathological assessment revealed that TDP-43 n/c-ir neurons became less dominant than c-ir neurons during prolonged disease durations. The fact indicates that the n/c-ir-dominant state is linked with a rapid progression of ALS. We speculate a possibility of high neurotoxicity demonstrated by poorly ubiquitylated, diffuse aggregates of TDP-43. Although studies have revealed that diffuse ‘preinclusions’ of TDP-43 lack amyloid nature detected by thioflavin S, an ultrastructural study clarified partial but definite formation of misfolded fibrils within the preinclusions [30]; at least, the diffuse TDP-43 preinclusions may have neurotoxicity as mature inclusions. It can be hypothesized that diffuse spreading of TDP-43 fibrils in the cytoplasm, avoiding ubiquitylation and following metabolization processes, demonstrates high neurotoxicity. Further basic studies needs to clarify which type or phase of TDP-43 inclusions is more neurotoxic than others.

The molecular basis of TDP-43 pathology includes many unclear components. An imbalance in the nuclear and cytoplasmic burden of TDP-43 has been hypothesized for the combination of cytoplasmic aggregation and nuclear clearance observed in ALS-TDP and FTLD-TDP. TDP-43 is actively and passively transported across the nuclear membrane in cells [32], and autoregulation within the nucleus is an important driver of TDP-43 expression levels [33]. The inhibition of nuclear localization factors of TDP-43, including nuclear localization signals and RNA-recognition motifs, results in the misfolding and prolonged half-life of TDP-43 [13,20,34]. If the hypothesis of TDP-43 imbalance is true for TDP-43 pathology, negative correlations between the homeostasis of nuclear TDP-43 and the development of cytoplasmic aggregation would be found in ALS cases. Our results from a comparison of TDP-43 c-ir and n/c-ir neurons suggested a link between the depletion of nuclear TDP-43 and the maturation of cytoplasmic aggregates, which support the hypothesis of TDP-43 imbalance between nucleus and cytoplasm.

TDP-43 is also abundantly expressed in the nuclei of glial cells. Glial cytoplasmic inclusion of TDP-43 is another finding in ALS-TDP and FTLD-TDP [26,27]. The inclusions are mostly observed in oligodendroglia, rather than astroglia, and are coiled, tricorn-like, ring-shaped, or dens punctate [11,26,27]. As neuronal inclusions, nuclear clearance of TDP-43 accompanies glial cytoplasmic inclusions [27]. Physiological TDP-43 has been reported to be critical for the survival and myelin formation of oligodendrocytes [35], although the importance of glial TDP-43 pathology is not fully understood in ALS or FTLD cases. Our study revealed that a subset of TDP-43 glial cytoplasmic inclusions was combined with the expression of nuclear TDP-43, termed n/c-ir glia. Interestingly, our clinicopathological assessment showed that n/c-ir glia were observed among cases with both short and prolonged clinical durations, whereas n/c-ir neurons were never found among cases with a disease duration of 49 months or longer. In previous reports, it was hypothesized that glial cytoplasmic inclusions of TDP-43 are consequent to neuronal inclusions via interactions between motor neuron axons and myelin sheaths [11,36]. A postmortem study addressing an ALS case who committed suicide five months after onset reported that preinclusions of TDP-43 were abundant in the neurons, but no inclusions were observed in the glia [36]. When taken together with our results, the development of glial cytoplasmic inclusions might occur in a later disease phase than the development of neuronal inclusions.

As a limitation, our study design was cross-sectional and retrospective. We evaluated the correlation between the maturation of TDP-43 aggregates and the reduction in nuclear TDP-43, but it remains uncertain which of the cytoplasmic and nuclear events causes the other. Another unsolved question is why there were fewer TDP-43 n/c-ir neurons in the primary motor cortex than in the lower motor neuron system. Studies have hypothesized that the propagation of TDP-43 aggregates begins from the primary motor cortex and then spreads to the lower motor neurons [11,12]. A possible explanation is that most TDP-43 n/c-ir, suggested to be a premature TDP-43 pathology, has already become less dominant than a mature TDP-43 pathology in the primary motor cortex because this portion is involved by the TDP-43 pathology earlier than in the lower motor neuron system. Another speculation is that processes of TDP-43 pathology are partially different between the upper and lower motor neuron systems. An autopsy-based study reported that in the upper motor neuron, nuclear TDP-43 often disappeared, but cytoplasmic aggregates were relatively rare; the study hypothesized that cytoplasmic TDP-43 in affected upper motor neuron might mostly remain in a soluble state, different from that in the lower motor neuron [37].

In conclusion, the morphology and ubiquitylation of cytoplasmic TDP-43 aggregates were clearly linked with the nuclear expression level of this protein; premature and poorly ubiquitylated TDP-43 inclusions were associated with nuclear expression, whereas mature and well-ubiquitylated inclusions were associated with nuclear clearance. Our results indicate that the pathogenesis of cytoplasmic aggregation progresses together with that of intranuclear reduction in TDP-43, which may be supportive of the hypothesis of TDP-43 imbalance inside and outside of the nucleus as being essential for TDP-43 pathology.

## 4. Materials and Methods

### 4.1. Cases

Signed informed consent for autopsy and tissue usage for research purposes was obtained from the family members of all cases, and research usage of autopsy-proven tissues was approved by the ethical committee of Aichi Medical University. We reviewed 26 pathologically diagnosed cases with ALS-TDP who were consecutively autopsied from Jan. 2015 to Aug. 2016 at Aichi Medical University. The pathological diagnosis of ALS-TDP was made according to the loss of upper and lower motor neurons in association with intraneuronal TDP-43 aggregations [1,2,3]. Clinical findings of all studied cases satisfied the possible or more severe categories in the revised El Escorial criteria [38]. We excluded 4 cases with inadequate tissue, and 22 cases were ultimately included in the study. Exclusion criteria were as follows: mechanical ventilation for more than a month (*n* = 2), comorbid neuropathological changes of Alzheimer’s disease that were classified into high grade [39] (*n* = 1), and comorbid neuropathological changes of limbic/diffuse neocortical stage of Lewy body disease [40] combined with Alzheimer’s disease (*n* = 1). We also evaluated 15 controls without any neurological disorders involving the spinal cord. The main disorders of the controls comprised muscle disease (*n* = 4), autoimmune disease (*n* = 4), and cardiovascular disease (*n* = 4), including normal pressure hydrocephalus (*n* = 1), cerebral contusion (*n* = 1), and chronic nephritis (*n* = 1). The age at death and sex of the controls were matched to those of the ALS cases.

### 4.2. Clinical Analyses

Clinical analyses are retrospective. Neurological findings, age at death, disease duration, and family history were obtained from clinical records.

### 4.3. Tissue Preparation

The left hemisphere of the brain and the spinal cord were fixed in 20% neutral-buffered formalin for at least two months, followed by paraffin embedding. The paraffin blocks were cut at a thickness of 5 μm for immunohistochemistry and 9 μm for hematoxylin–eosin and Klüver–Barrera methods.

Segments of the spinal cord were identified with a classic method before cutting; the anterior roots became much thinner in the S2 segment than in the S1 segment or more rostral. The dorsal roots become much thinner in T2 than in T1 or more rostral [41,42]. Then, we cut the spinal cord at the most rostral point where the anterior roots exit the cord parenchyma (Appendix A). The C4-8 of cervical segments, T1-6 of thoracic segments, L1-5 of lumbar segments, and S2 of the sacral segment were evaluated in the study. Some of these segments were lacking in a subset of cases (Appendix A) because they were subjected to frozen preservation. The medulla oblongata was axially cut at 5 mm caudal from the obex. The primary motor cortex was obtained from a coronal section at the level of the thalamic pulvinar.

### 4.4. Immunohistochemistry

Anti-full-length TDP-43 (panTDP-43) immunohistochemistry was used to label intrinsic nuclear TDP-43 and pathological aggregates. Anti-phosphorylated TDP-43 (p-TDP-43) immunohistochemistry recognized aggregated TDP-43 but not intrinsic nuclear TDP-43. The primary antibodies that we used were as follows: anti-alpha-synuclein (rabbit polyclonal, 1:2000, Sigma-Aldrich, St. Louis, MO, USA), anti-beta-amyloid (12B2, mouse monoclonal, 1:500, Immuno-Biological Laboratories, Minneapolis, MN, USA), anti-cystatin-C (mouse monoclonal, 1:500, Santa Cruz, Dallas, TX, USA), anti-hyperphosphorylated tau (AT8, mouse monoclonal, 1:3000, Thermo Fisher Scientific, Waltham, MA, USA), anti-p62 (LCK-ligand, mouse monoclonal, 1:500, BD Biosciences, San Jose, CA, USA), anti-panTDP-43 antibody (rabbit polyclonal, 1:2500, Proteintech, Chicago, IL, USA), anti-phosphorylated TDP-43 (p-TDP-43 s409/410, mouse monoclonal, 1:3000, Cosmobio, Tokyo, Japan), and anti-ubiquitin (ubi-1, mouse monoclonal, 1:1000, EMD-Millipore, Burlington, MA, USA) antibodies. Secondary immunolabeling was performed using biotin-tagged immunoglobulins via the standard avidin–biotin method. We used 3,3′-diaminobenzidine (Wako) as the chromogen.

### 4.5. Immunofluorescence

Secondary antibodies were conjugated with Alexa Fluor 405, 488, or 568 (Molecular Probes, Carlsbad, CA, USA). 4,6-Diamidino-2-phenylindole (DAPI) was used to stain nuclei. The labeled specimens were observed with a laser confocal microscope (FV3000, Olympus, Tokyo, Japan). Images were acquired under the same laser and detection settings for each combination of primary antibodies.

### 4.6. Quantitative Analyses

We counted neurons with cytoplasmic TDP-43 inclusions in the primary motor cortex, hypoglossal nerve nucleus, and spinal cord anterior horn. We excluded neurons for which nuclei were not present in specimens because the study aimed to assess nuclear immunostaining of TDP-43. For the hypoglossal nerve nucleus and spinal cord anterior horn, the neurons were counted in the whole visual field of both sides. In contrast, for the primary motor cortex, we counted these in five randomly chosen visual fields with distances of 500 μm between each other. Counting was performed under ×200 magnification with a 3.8 mm^2^ visual field.

In this study, we defined the anterior horn of the spinal cord as follows: the anterior border of the anterior horn was defined by the boundary between the gray matter and anterior column, whereas the dorsal border of the anterior horn was defined by a line running between the most lateral point of the anterior gray matter and the most dorsal point of the medial anterior column (Appendix A).

### 4.7. Identification of TDP-43 c-ir and n/c-ir

Neurons or glial cells with cytoplasmic TDP-43 aggregation were subclassified into two groups based on the expression or clearance of nuclear TDP-43. On the basis of DAB-labeled anti-panTDP-43 immunohistochemistry with hematoxylin counterstaining, we defined cytoplasmic TDP-43 immunoreactivity in combination with complete nuclear clearance as ‘TDP-43 c-ir (cytoplasmic immunoreactivity)’; in this setting, DAB-staining disappeared from the nucleus, and only hematoxylin staining was visible within the nucleus. In contrast, cytoplasmic immunoreactivity in association with nuclear expression of TDP-43 was defined as ‘TDP-43 n/c-ir (nuclear and cytoplasmic immunoreactivity)’; in this setting, TDP-43-immunostaining can be observed within the nucleus at all expression levels, ranging from almost normally dense staining to weak staining. The judgment of TDP-43 c-ir or n/c-ir was made by three observers (H. Y, Y.R, and M.Y). Nonspecific TDP-43 immunostaining, which shows coarse granules, amorphous staining, and lipofuscin staining (Appendix A), was excluded from counts.

### 4.8. Classification of Neuronal TDP-43 Aggregates

For lower motor neurons, cytoplasmic inclusion bodies of TDP-43 were classified into diffuse, skein-like, and round (Appendix A). FTLD-related TDP-43 inclusions in the cerebral cortices were classified into types A, B, or C using published harmonized pathological criteria [28]; type A is characterized by short dystrophic neurites and small, dense cytoplasmic inclusions in the superficial cortical layers; type B shows diffuse or ring-shaped neuronal inclusions across all cortical layers; and type C is characterized by long dystrophic neurites in the superficial cortical layers with sparse cytoplasmic inclusions.

### 4.9. Colocalization Assay

Using double immunofluorescent slides, we computed the colocalization rate as (area of colocalized part (labeled yellow)/area of cytoplasmic inclusion (labeled red)). The computation was performed for each neuron. In brief, photomicrographs were captured under ×1000 magnification using a laser confocal microscope in the same settings of gain, pinhole value, photomultiplier voltage, and resolution. We automatically detected fluorescent signals using the Otsu-thresholding filter of ImageJ (NIH, Bethesda, MD, USA) for the red, green, and blue channels. Colocalized areas with yellow signals in merged images were also automatically detected using the AND-operation of the paste function of ImageJ (Appendix A).

### 4.10. Measurement of Nuclear Signal Intensity

Using immunofluorescent slides, we measured fluorescent signal intensity in the neuronal nucleus. The nuclei were identified with DAPI labeling. Photomicrographs were captured under ×1000 magnification using a laser confocal microscope in the same setting. We generated a stacked image for each slide, and the measurement was undertaken at the point where the nucleus most largely appeared. Signal intensity along the longest diameter running neuronal nucleus was measured using the plot profile function of ImageJ. In this method, a value of 100 means the highest (saturated) intensity, whereas 0 means the lowest (no signal) intensity. Values obtained from pixels along the diameter were averaged, and the averaged values were defined as the nuclear signal intensity of a given neuron (Appendix A).

### 4.11. Statistical Analysis

The clinical and pathological findings were compared using the Mann-Whitney U test for quantitative variables and Fisher’s exact test for qualitative variables. We employed Spearman’s rank order to assess the correlations of nonparametric variables. The significance level was set at 0.05 for comparisons between two groups. Bonferroni correction was performed for *p* values obtained from multiple comparisons. All statistical analyses were two-sided and were performed using the SPSS 21.0 software program (IBM, Armonk, NY, USA).

## Figures and Tables

**Figure 1 ijms-24-12176-f001:**
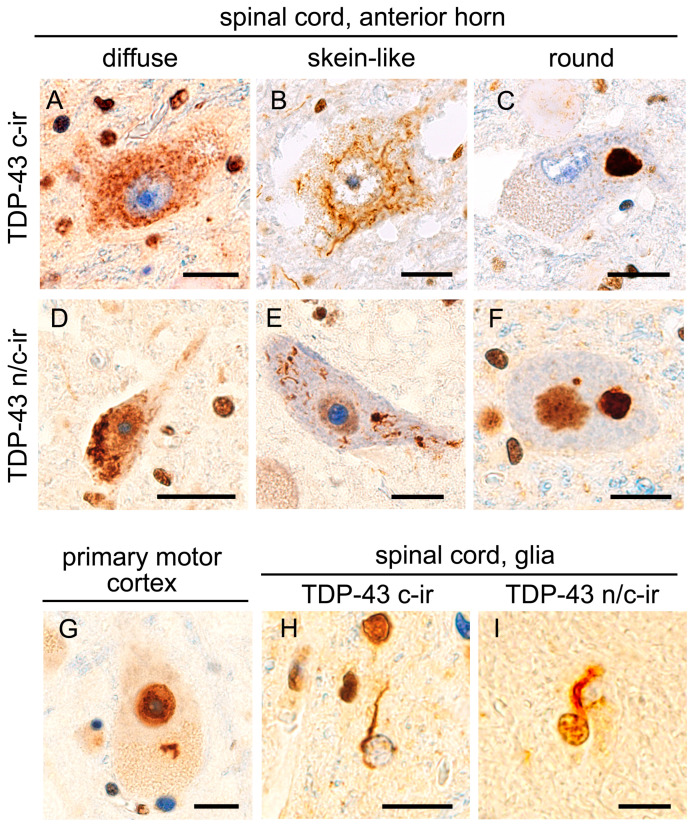
TDP-43 aggregation with and without nuclear clearance. All panels show anti-full-length TDP-43 (panTDP-43) immunohistochemistry of studied cases. TDP-43 pathology in ALS cases was characterized by diffuse (**A**), skein-like (**B**), or round (**C**) cytoplasmic inclusions in combination with the clearance of intrinsic, nuclear TDP-43 (TDP-43 c-ir, (**A**–**C**)). However, cytoplasmic inclusions occasionally arose in neurons showing nuclear TDP-43 expression (TDP-43 n/c-ir, (**D**–**F**)). TDP-43 n/c-ir was also observed in Betz cells in the primary motor cortex (**G**). Although TDP-43 aggregates of the oligodendroglia were mostly combined with nuclear clearance (**H**), a subset showed immunopositivity in the nucleus (**I**). Bars = 10 μm.

**Figure 2 ijms-24-12176-f002:**
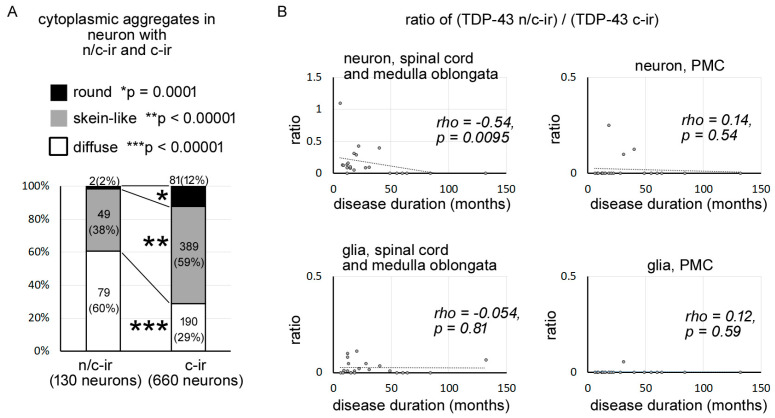
TDP-43 n/c-ir was associated with diffuse cytoplasmic inclusions and became less dominant in prolonged clinical durations. (**A**) In total, we identified 790 neurons (n/c-ir neurons = 130, c-ir neurons = 660) with TDP-43 aggregates from the medulla oblongata and spinal cord of all studied cases with ALS-TDP. Diffuse inclusions were more prevalent in TDP-43 n/c-ir neurons than in TDP-43 c-ir neurons (*p* < 0.00001, Fisher’s exact test, significance level was set at 0.016 after Bonferroni correction), whereas skein-like and round inclusions were more prevalent in TDP-43 c-ir neurons (skein-like inclusions *p* < 0.0001; round inclusions *p* = 0.0001). Neuron counts (%) in each inclusion type are shown in graph bars. (*), (**), and (***) indicate significant differences in round inclusion, skein-like inclusion, and diffuse inclusion, respectively. (**B**) The ratio ((TDP-43 n/c-ir neuron)/(TDP-43 c-ir neurons)) was negatively correlated with the disease duration of ALS (rho = −0.54, *p* = 0.0095, Spearman’s rank order) in the lower motor neuron system. Cases with a clinical duration of 49 months or longer never showed TDP-43 n/c-ir neurons, whereas a subset of these cases had TDP-43 n/c-ir glia in the spinal cord or medulla oblongata.

**Figure 3 ijms-24-12176-f003:**
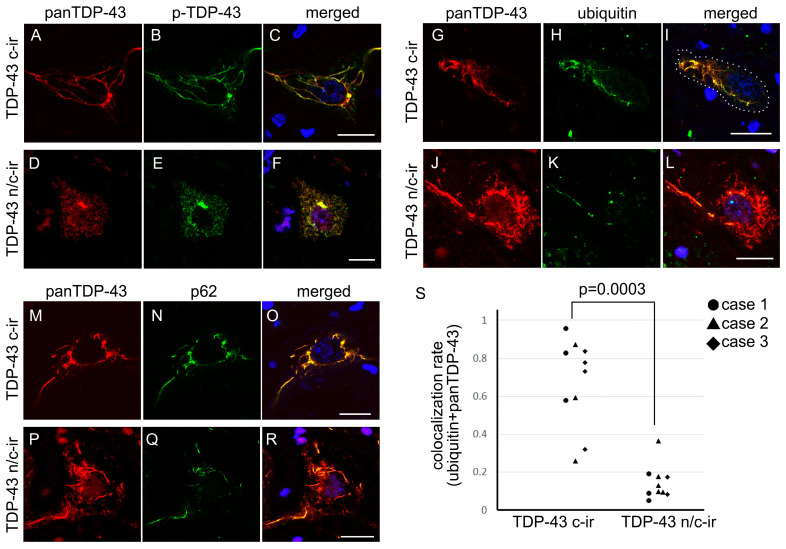
Cytoplasmic inclusions of TDP-43 n/c-ir neurons were poorly ubiquitylated. All panels were taken from the spinal cord anterior horn. The cytoplasmic panTDP-43 signal colocalized with phosphorylated TDP-43 (p-TDP43) in c-ir neurons (**A**–**C**) and n/c-ir neurons (**D**–**F**). Cytoplasmic panTDP-43 signals were mostly colocalized with those of ubiquitin or p62 in TDP-43 c-ir neurons (**G**–**I**,**M**–**O**); however, in TDP-43 n/c-ir neurons, cytoplasmic inclusions were partially colocalized with ubiquitin or p62 (**P**–**R**,**J**–**L**). A colocalization assay of skein-like inclusions demonstrated a significantly lower rate in TDP-43 n/c-ir neurons (mean 0.14 ± 0.09) than in c-ir neurons (mean 0.67 ± 0.23) (*p* = 0.0003, Mann-Whitney U test) for panTDP-43 and ubiquitin (**S**). Scale bars = 10 μm.

**Figure 4 ijms-24-12176-f004:**
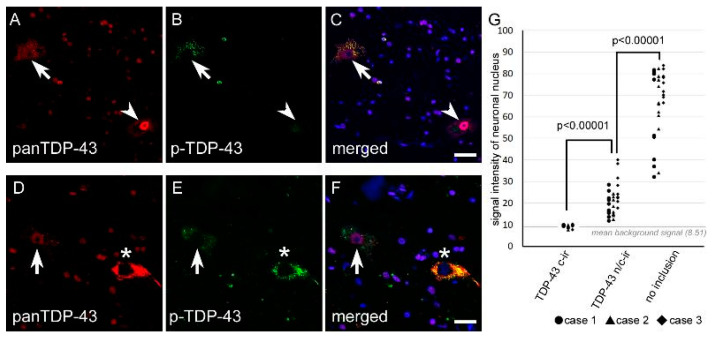
The signal intensity of nuclear TDP-43 was lower in n/c-ir neurons than in normal neurons. All panels were taken from the spinal cord anterior horn. (**A**–**C**) Nuclear TDP-43 immunostaining of a n/c-ir neuron (arrows) and a normal neuron without inclusions (arrowheads) are shown. (**D**–**F**) These panels show nuclear TDP-43 immunostaining of a n/c-ir neuron (arrows) and a c-ir neuron (asterisks). (**G**) TDP-43 n/c-ir neurons showed significantly lower signal intensity of nuclear TDP-43 (mean 21.12 ± 7.18) than normal neurons (mean 67.38 ± 15.88) (*p* < 0.00001, Mann-Whitney U test). The nuclear signal intensity of TDP-43 c-ir (mean 9.47 ± 0.57) was almost the same as the averaged background intensity shown with a gray line (mean 8.51 ± 0.16). Scale bars = 20 μm.

**Figure 5 ijms-24-12176-f005:**
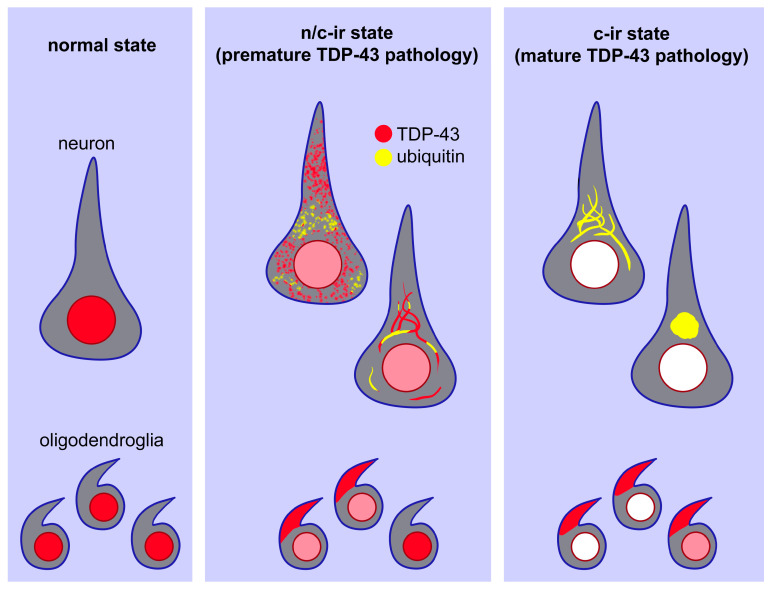
A scheme of the hypothesized process of TDP-43 pathology. In a normal setting, TDP-43 is localized in the neuronal and oligodendroglial nuclei (**left panel**). In a transitional state, TDP-43 aggregates in the cytoplasm in association with partial ubiquitylation. TDP-43 remains in the nucleus, but its expression level is lower than that in the normal state (**middle panel**). At a mature state of TDP-43 pathology, TDP-43 aggregates become more consolidated and are fully ubiquitylated. Nuclear TDP-43 disappears in neurons but remains in some oligodendroglia (**right panel**).

**Table 1 ijms-24-12176-t001:** Demographic characteristics of the cases included.

	ALS	Control	*p*
*n* (female/male)	22 (8f/14m)	15 (3f/12m)	0.466 *
age at death (mean, SD)	62.7 ± 10.1	69.7 ± 10.2	0.529 **
duration of disease (month, median, range)	19 (6-132)	-	
postmortem delay (hour, median, range)	3 (1.5 − 24)	9 (3.5 − 20)	0.200 **
brain weight	1298 ± 120	1260 ± 204	0.267 **
brain aging pathology			
Braak’s tau stage (mean, SD, range)	1.5 ± 0.7 (I–III)	1.7 ± 0.9 (0–III)	0.447 **
Lewy body disease (mean, SD, range)	0	0	1
Thal’s amyloid phase (mean, SD, range)	0.7 ± 0.7 (0–2)	0.7 ± 0.8 (0–2)	0.984 **

* Fisher’s exact test, ** Mann–Whitney U-test. ALS amyotrophic lateral sclerosis, SD standard deviation.

## Data Availability

The data is secured by authors’ institute because this study is based on patient-derived material. Anonymized data can partially be provided with contacting Y.R. (riku.yuuichi.498@mail.aichi-med-u.ac.jp).

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
