# Peer review of "Nuclear Expression of TDP-43 Is Linked with Morphology and Ubiquitylation of Cytoplasmic Aggregates in Amyotrophic Lateral Sclerosis"

_ijms, 2023, doi:10.3390/ijms241512176_

Round 1

Reviewer 1 Report

In this paper, the authors state that the distribution of TDP-43 in neurons moves from the nucleus to the cytoplasm and forms inclusion bodies as the disease progresses.  They describe that quantitative balance of nuclear and cytoplasmic TDP-43 is important, and that loss of TDP-43 in the nucleus leads to disease, which is the basic pathological picture.  The reviewer have no objection to the author's views.  However, the reviewers would like clarification on the data presented by the authors.

  1. There are more TDP-43-positive images in glial cells than in neurons, so what is the reason for focusing on neurons?
  2. Show the relationship between the author's idea of maturation of TDP-43 inclusion bodies and cell survival.
  3. Why does maturation of TDP-43 inclusion bodies not occur in glial cells?  The reviewer would like to describe the author's opinion.
  4. What is the cause of the TDP-43 cytoplasmic translocation that the author thinks?

Author Response

Reviewer 1:

1) There are more TDP-43-positive images in glial cells than in neurons, so what is the reason for focusing on neurons?

We appreciate the deep insight. As the referee commented, we undertook quantification of neuronal TDP-43 pathology but not glial one. There are two reasons. The first, implications of glial TDP-43 pathology/physiology may be controversial, although neuronal TDP-43 pathology/physiology can directly be connected to motor function and motor neuron surviving; for instance, a subset of ALS cases is lacking glial TDP-43 pathology as discussed in the lines 323-325. The second is a technical reason. Glial TDP-43 pathology is mostly in oligodendroglial origin, but antibodies or staining methods excellently labeling oligodendroglia are not available. Hence, precise quantifications of TDP-43 n/c-ir or c-ir glia were hard to address, which made a contrast with the setting of neurons. We also had attempted to assess the spinal cord white matters, however, TDP-43 pathology was unexpectantly sparse there; glial TDP-43 aggregation is often nearby the neurons with TDP-43 aggregations in the gray matter (Brettschneider J, et al. Ann Neurol 2014).

2) Show the relationship between the author's idea of maturation of TDP-43 inclusion bodies and cell survival.

It is an important aspect, and we have added some writings about reported correlations between maturation of TDP-43 pathology and biological outcome or neuronal counts (the lines 276-278). Our study is cross-sectional, hence effects of maturation of TDP-43 pathology toward cell surviving cannot be addressed directly. For autopsy-based designs, at least, studies (i.e., references [23] by Giordana, et al. and [30] by Mori, et al.) clarified correlation between maturation of TDP-43 inclusions and neuronal counts or clinical durations. This issue could be addressed using animal/cell models, although it is difficult to completely reproduce TDP-43 inclusions that include skein-like inclusions and round inclusions as seen in human.

3) Why does maturation of TDP-43 inclusion bodies not occur in glial cells?  The reviewer would like to describe the author's opinion.

The biggest reason is that glial TDP-43 inclusions are poorly ubiquitylated as reported by Arai, et al. [26] and Brandmeir, et al. [27]. It was true in our cases, regardless of n/c-ir or c-ir. Hence, we cannot apply the same strategy as neurons toward a maturation assay of glial TDP-43 aggregation. The reason for poor ubiquitylation of glial TDP-43 inclusions is unknown, although we have a speculation that is shown in the lines 319-329.

4) What is the cause of the TDP-43 cytoplasmic translocation that the author thinks?

It is the biggest and most important controversy. Largely, there may be two aspects. One is propagation of aggregative seed from other cells. Another is imbalance of nuclear and cytosolic fractions of TDP-43. Our study tends to stand on the latter hypothesis because reduction of nuclear TDP-43 correlated with maturation of cytosolic TDP-43 aggregates. Physiological background that initiates ‘imbalance’ is unknown and is often abstracted as ’cellular/protein aging’, although recent studies have clarified detailed factors relevant to trans-nuclear-membrane transport of TDP-43 (Tziortzouda P, et al. Nat Neurosi 2022;22:197-208, review). Interestingly, our previous study revealed that induction of TDP-43 aggregative seed into ALS patient-derived iPS organoid resulted in nuclear clearance of TDP-43, and control-derived iPS did not show it (Tamaki, et al. [22]). Currently, we speculate that ALS patients fundamentally have a nature of TDP-43 imbalance (i.e., dysfunctions in trans-nuclear-membrane transport of TDP-43), and cytoplasmic aggregation seeds facilitate the imbalance and mislocalization.

Reviewer 2 Report

Present research article by Yabata et al. entitled "Nuclear expression of TDP-43 is related to morphology and ubiquitylation of cytoplasmic aggregates" demonstrates the qualitative assessment of TDP-43 inclusions in cytoplasm of brain tissues obtained from sporadic ALS subjects. Authors have nicely correlated the nuclear expression of TDP-43 and its cytoplasmic distribution in advanced pathological conditions. There are no specific queries. Just a curiosity, did authors also analyzed the brain tissues quantitatively? e.g. by western blotting of nuclear vs cytosolic fractions. 

English language is fine.

Author Response

Reviewer 2:

Present research article by Yabata et al. entitled "Nuclear expression of TDP-43 is related to morphology and ubiquitylation of cytoplasmic aggregates" demonstrates the qualitative assessment of TDP-43 inclusions in cytoplasm of brain tissues obtained from sporadic ALS subjects. Authors have nicely correlated the nuclear expression of TDP-43 and its cytoplasmic distribution in advanced pathological conditions. There are no specific queries. Just a curiosity, did authors also analyzed the brain tissues quantitatively? e.g. by western blotting of nuclear vs cytosolic fractions.

Answer) We deeply appreciate the attractive insight. We also feel extraction of nuclear and cytoplasmic fraction to give further information. During study period, we had attempted to extract the nuclear fraction from frozen brain and spinal cord. However, it is technically known that extraction is much difficult in frozen CNS tissue than in raw cultured cells or in non-neural organs, and we could not achieve complete removal of cytosolic fraction from the nuclear fraction. Therefore, experiments using nuclear fraction from frozen brain/cord are currently hard to achieve. Conversely, basic studies have tried to overcome this issue (Kuster DWD, et al. J Physiol. Biochem. 67:165-173), hence, we will address this problem again in future studies.

Reviewer 3 Report

TDP-43 (TAR DNA-binding protein 43) is a protein that plays a crucial role in RNA processing and regulation in the cell nucleus. In certain neurodegenerative disorders, such as amyotrophic lateral sclerosis (ALS) and frontotemporal lobar degeneration (FTLD), TDP-43 becomes mislocalized and forms cytoplasmic aggregates, which are a hallmark feature of these diseases. Importantly, very recent study demonstrated that RNA-binding domain mutations markedly reduce TDP-43 nuclear localization and abolish transcription blockade-induced nuclear efflux (PMID: 35858577)

The morphology of TDP-43 aggregates can vary depending on the disease and the stage of the pathology. The aggregates can be found in neurons and glial cells, and their presence is associated with cellular dysfunction and neurodegeneration. Earlier studies have shown that TDP-43 aggregates are often associated with ubiquitin molecules, indicating that ubiquitylation is involved in their formation or clearance. Another study reported that Hyper-phosphorylated and ubiquitinated TDP-43 deposits could act as inclusion bodies in the brain and spinal cord of patients with the motor neuron diseases: amyotrophic lateral sclerosis (ALS) and frontotemporal lobar degeneration (FTLD) (PMID: 30837838). However, the most common tau-negative variant is FTLD with ubiquitin-immunoreactive lesions (FTLD-U) and TDP-43 was identified in neuronal inclusions in FTLD-U. Case study indicated that after applying TDP-43 immunohistochemistry to a series of 44 cases of FTLD-U, only three cases (7%) were identified with ubiquitin- and p62-positive neuronal cytoplasmic inclusions (NCI) that were negative for TDP-43 (PMID: 18553091). Furthermore, Case study reported that pTDP-43 (phosphorylated TDP-43) levels were significantly higher in ALS patients (PMID: 34194383), In contrary to this, another recent study showed that C-terminal TDP-43 phosphorylation as detected on ALS/FTD inclusions has an inhibitory effect on TDP-43 aggregation, underscoring the idea that aberrant PTMs detected on pathological inclusions may not necessarily all be drivers of protein aggregation, but could also have protective, anti-aggregation effects (PMID: 35112738).

In summary, TDP-43 is a protein that becomes mislocalized and forms cytoplasmic aggregates in neurodegenerative disorders such as ALS and FTLD. The morphology of these aggregates can vary and is associated with cellular dysfunction. Ubiquitylation or phosphorylation, a post-translational modification, whether are involved in the formation and clearance of TDP-43 aggregates are still not clear based on previous study.

Here authors aimed to identify whether depletion of nuclear TDP-43 is linked to the properties of TDP-43 cytoplasmic aggregates TDP-43 based on cytoplasmic immunoreactivity (c-ir) or nuclear and cytoplasmic immunoreactivity (n/c-ir) and morphology and ubiquitylation of cytoplasmic TDP-43 aggregates. Indeed, the questions here authors tried to address relevant in TDP-43 biology and also indicting the novel prospective of the study. However, I offered some suggestions for how the authors might be able to improve the manuscript, some of the points are mentioned as follows: 

1) In Line 107-110, authors mentioned TDP-43 immunoreactivity as c-ir or n/c-ir is not very clear. Authors should elaborate briefly the Materials and methos section “Identification of TDP-43 c-ir and n/c-ir” on this how the immunoreactivity identified based on DAB-labeled, anti-panTDP-43 immunohistochemistry with hematoxylin counterstaining, as this is one of the major aspect of the study.

2) In line 281-284, authors mentioned “The poor ubiquitylation suggests that cytoplasmic TDP-43 aggregates of n/c-ir neurons are mostly in a premature state of aggregation. Moreover, the signal intensity of nuclear TDP-43 was significantly lower in n/c-ir neurons than in normal neurons, indicating that nuclear expression level of TDP-43 may partially reduce at this state.” Authors need to add references supporting this statement and discuss it more briefly.

3) In Fig 3(D-F), p-TDP-43 fraction colocalized with cytoplasmic fraction while using panTDP-43 in n/c-ir neuron. However, this is interesting observation and authors need to discuss briefly to reflect the relevance of the experiment and role of p-TDP-43 in pathogenicity.

4) In Figure 2, author could add percentage (as number) of round, skein-like, diffuse aggregate in n/c-ir and c-ir neurons for clear representation of the graph.

5) In Figure 3, the graph ‘S’ label is missing in the legend in line 221.

6) In graph 3S, 4G, authors need to mention the mean values in the corresponding legend or mark in the graph.

Author Response

1) In Line 107-110, authors mentioned TDP-43 immunoreactivity as c-ir or n/c-ir is not very clear. Authors should elaborate briefly the Materials and methos section “Identification of TDP-43 c-ir and n/c-ir” on this how the immunoreactivity identified based on DAB-labeled, anti-panTDP-43 immunohistochemistry with hematoxylin counterstaining, as this is one of the major aspect of the study.

Answer) We improved the definition of c-ir and n/c-ir in the M&M section. We also guess that our former writing ‘sparing nuclear TDP-43’ made confusion because it was unclear if sparing indicated ‘sparing in normal level’ or ‘expressing in any levels, including normally spared and partially reduced’. We have corrected this point throughout the text.

2) In line 281-284, authors mentioned “The poor ubiquitylation suggests that cytoplasmic TDP-43 aggregates of n/c-ir neurons are mostly in a premature state of aggregation. Moreover, the signal intensity of nuclear TDP-43 was significantly lower in n/c-ir neurons than in normal neurons, indicating that nuclear expression level of TDP-43 may partially reduce at this state.” Authors need to add references supporting this statement and discuss it more briefly.

Answer) In autopsy based studies, at least, it is reported that TDP-43 aggregations spread more broadly in the brain than ubiquitin inclusions. It is also known that round or skein-like inclusions are more ubiquitylated than diffuse, preinclusions. Although several studies have investigated this point, we consider that a paper by Giordana MT, et al. (Brain Pathol 2010) comprehensively and concisely addressed relations between ubiquitin and maturation of TDP-43 aggregation. Thus, we improved the writing of given portion and added the literature there.

3) In Fig 3(D-F), p-TDP-43 fraction colocalized with cytoplasmic fraction while using panTDP-43 in n/c-ir neuron. However, this is interesting observation and authors need to discuss briefly to reflect the relevance of the experiment and role of p-TDP-43 in pathogenicity.

Answer) As the referee noted, this study stands on the point that anti-phosphorylated TDP-43 immunohistochemistry is the reliable marker of aggregative TDP-43. An early paper by Hasegawa M, et al. illustratively clarified correlations between CK1-induced hyper-phosphorylation and aggregation of TDP-43 or between hyper-phosphorylation and insoluble CTF (Ann Neurol 2008). The former correlation (CK1-experiment) may suggest causality of phosphorylation toward an aggregative nature, although the pathway contains many unclear parts. In the revision, we improved writing of Discussion on the basis of these reported facts (the lines 254-268). Currently, molecular and biochemical causality from phosphorylation toward misfolding of this protein is precisely unclear and is hardly approached by methodologies in our current study.

4) In Figure 2, author could add percentage (as number) of round, skein-like, diffuse aggregate in n/c-ir and c-ir neurons for clear representation of the graph.

Answer) We appreciate this suggestion to improve display. The percentage and number have been added into the figure panel.

5) In Figure 3, the graph ‘S’ label is missing in the legend in line 221.

Answer) We have added the label for Figure 3S.

6) In graph 3S, 4G, authors need to mention the mean values in the corresponding legend or mark in the graph.

Answer) We have added the mean (SD) into the figure legend.